# The Clinical Validation of Modulated Electro-Hyperthermia (mEHT)

**DOI:** 10.3390/cancers15184569

**Published:** 2023-09-15

**Authors:** Sun-Young Lee, Gergo Lorant, Laszlo Grand, Attila Marcell Szasz

**Affiliations:** 1Department of Radiation Oncology, Jeonbuk National University Medical School, Jeonju 54907, Republic of Korea; sylee78@jbnu.ac.kr; 2Research Institute of Clinical Medicine of Jeonbuk National University-Biomedical Research Institute of Jeonbuk National University Hospital, Jeonju 54907, Republic of Korea; 3Division of Oncology, Department of Internal Medicine and Oncology, Semmelweis University, H-1083 Budapest, Hungary; lorant.gergo@phd.semmelweis.hu; 4Faculty of Information Technology and Bionics, Pázmány Péter Catholic University, H-1083 Budapest, Hungary; grand.laszlo.balint@itk.ppke.hu

**Keywords:** heterogenic heating, cellular selection, thermal processes, nonthermal actions, clinical studies, clinical evidence, survival time, quality of life

## Abstract

**Simple Summary:**

Modulated electro-hyperthermia (mEHT) is a heating therapy that uses synergized thermal and nonthermal effects to heat and destroy malignant cells selectively without damaging healthy cells. This article presents the clinical validation of mEHT. The therapy is dominantly applied for such advanced malignancies when the conventional oncotherapies fail to apply. Survival results of mEHT were collected and compared with other methods. The results demonstrate the superiority of the mEHT method.

**Abstract:**

The mEHT method uses tissues’ thermal and bioelectromagnetic heterogeneity for the selective mechanisms. The success of the therapy for advanced, relapsed, and metastatic aggressive tumors can only be demonstrated by measuring survival time and quality of life (QoL). The complication is that mEHT-treated patients cannot be curatively treated any longer with “gold standards”, where the permanent progression of the disease, the refractory, relapsing situation, the organ failure, the worsening of blood counts, etc., block them. Collecting a cohort of these patients is frequently impossible. Only an intent-to-treat (ITT) patient group was available. Due to the above limitations, many studies have single-arm data collection. The Phase III trial of advanced cervix tumors subgrouping of HIV-negative and -positive patients showed the stable efficacy of mEHT in all patients’ subgroups. The single-arm represents lower-level evidence, which can be improved by comparing the survival data of various studies from different institutes. The Kaplan–Meier probability comparison had no significant differences, so pooled data were compared to other methods. Following this approach, we demonstrate the feasibility and superiority of mEHT in the cases of glioblastoma multiform, pancreas carcinomas, lung tumors, and colorectal tumors.

## 1. Introduction

The history of clinical hyperthermia can be traced back to the past. Ancient Greek medicine already used the method to treat oncological cases without detailed knowledge about the physiological feedback of the human body. Knowing the nature of the febrile condition, which in many cases was the guarantee of recovery, it was believed that proper heating would solve most medical problems [1]. They trusted that the body’s reactions would lead to the heating effect, inducing self-healing reactions of the organism. They essentially took advantage of the system’s striving for homeostatic balance, which is facilitated by heat stimuli. This principle is still appropriate in modern medicine, considering the limits of complex natural regulation.

Homeostatic surveillance controls the system’s stability and adaptability. The local or systemic heating interrupts the regulatory processes of stability, reacting with non-linear physiological responses to correct the inconsistency [2]. Theoretical biology often ignores this complex control facing a tragicomedy challenge [3]. The homeostatic control tries to re-establish the unheated conditions by non-linear feedback, increasing the cooling blood flow (BF) [4,5] as an effective heat exchanger. This complex dynamic behavior guarantees the robust stability of health conditions, so the reactive BF challenges the heating processes. Homeostatic control tries to restore healthy regulation by increasing blood flow and vasodilatation. However, the risk of invasion of tumor cells is enhanced by these corrective effects and may promote malignant dissemination.

Modulated electro-hyperthermia (mEHT) aims at a harmonic solution to this contradiction [6]. It applies electromagnetic interactions to deliver energy to the tumor. The energy is realized in the synergy of two basic effects:Thermal effects occur in the form of heat and temperature increase. Thermal effects are mostly unselective; the heat spreads all over the volume seeking thermal equilibrium. The temperature characterizes the homogeneous distribution as average energy of the heat-absorbers.Nonthermal processes are electron excitations, generating chemical reactions. The nonthermal impact may change the intercellular membrane, and intracellular processes select them by the dielectric and conductive heterogeneity of the target.

The mEHT applies a precise, personalized theranostic selection and treatment of malignancy, supporting natural homeostatic processes such as apoptosis, immune reactions, conditional effects, etc. [7]. The selection of the malignant cells uses the microscopic natural heterogeneities of the tumor. The applied electric field has different interactions with the cancerous and healthy cells in four basic characteristics, Figure 1, discussed in multiple publications [8,9,10,11,12]:Due to the intensive metabolism of the malignant cells, the ionic species of the nutrients and waste molecules (such as lactate) have high concentrations in the tumor microenvironment (TME), and together with the extended volume of the extracellular matrix (ECM), create a significantly higher electric conductivity of the microenvironment of malignant cells and the entire tumor [13,14]. This conduction difference drives the RF current to the area [15].The malignant cells break their networking connections (e.g., adherent connections and junctions [16]), and became autonomic. This cellular individualism makes the tumor microenvironment different, causing a higher dielectric permittivity of the tumor microenvironment (TME) than it was in the networking conditions [17,18]. The high dielectric permittivity favors conducting the radiofrequency (RF), making an additional selective factor for tumor cells.The broken connections leave numerous transmembrane proteins on the membrane of the malignant cells. These membrane-embedded proteins and their lipid-enriched clusters (membrane rafts) have significantly higher energy absorption from the RF current than their surrounding lipid layer [19]. This makes these proteins particularly heatable and chemically excitable.The malignancy had lost its healthy homeostatic control, and so it has locally modified physiologic regulations [20]. The arising structural and pathological modifications appear as an additional selectivity factor.

The auto-selection is theranostic, finding and treating the malignancy in macro- and micro-regions. The theranostic impact has special enhancing factors:The particular energy intake of membrane rafts of malignant cells selectively heats them, working similar to natural absorbing nanoparticles [21]. This makes effective micro- and macro-localization of the heat effect [22].The frequency dispersion has an optimal range of RF application for the above selection. However, the requested optimal frequency range of the membrane energy absorption/excitation and the driving of the molecular changes during the excitation need different frequencies which must be coordinated. The selection absorption optimum is near 10 MHz [23], while the desired molecular changes happen with a frequency less than 10 kHz. This 1/1000 ratio may be solved by modulation. The carrier is the approved medical frequency 13.56 MHz, and the modulation is a spectrum in the 10 Hz–10 kHz region [24]. The modulation spectrum is the physiologic noise of healthy homeostasis (its power density depends on the reciprocal value of the frequency) [25], and so forces the homeostatic control.The applied modulated RF current kills the malignant cells in an apoptotic way, producing a damage-associated molecular pattern (DAMP) [26], realizing an immunogenic cell death (ICD) [27]. The ICD secrete calreticulin (CRT) [28] and heat-shock protein (HSP) on the malignant membrane [29] and attract the natural killer cells [30]; this is proven with mEHT too [31]. The ICD liberates the high mobility group box 1 (HMGB1) molecules [32] together with HSP70, HSP90 [33], and ATP [34] into the ECM. The membrane HSP-s activate the natural killer cells, and the other DAMP molecules maturate the dendritic cells, producing antigen presentation which creates immune reactions. The rising tumor-specific killer and helper T-cells activate antitumoral processes all over the body, acting on distant micro- and macro-metastases (abscopal effect) [35].

The preclinically proven selective mEHT processes [36] have numerous clinical studies [37]. The clinical efficacy of these trials is focused on patient-centered values: survival time and quality of life.

A broad spectrum of cancers shows the practical applicability of mEHT in human oncotherapies [37], which validates the preclinical molecular results [36] and supports the method’s feasibility [9]. The applied low incident energy is enough to detect, select, and treat the tumors in a theranostic way, irrespective of their form and size, and reach significant clinical achievements through the selected cellular heating [38].

The mEHT application in human clinical practice showed typical thermally enhanced BF measured in cervical carcinoma [39]. The produced mild 38.5℃ temperature in cervical cancer looks optimal for the complementary treatments because it provides enough blood support but is not too high to increase the dissemination of the malignant cells. Furthermore, the synergy of the thermal and nonthermal processes improves the pharmacokinetics measured in healthy volunteers [40,41].

## 2. Clinical Validation

### 2.1. Crossroads of Clinical Applications

The general cancer curative strategy is based on the direct distortion of the detected cancer cells by surgery, chemo, and radiotherapy. The distortion strategy offers a proper solution when the method is effective and selective enough. Selectivity means that it destroys all cancer cells, does not dangerously affect its healthy neighborhood, and does not escalate to paralyze essential body functions. The accuracy of the detection of malignant cells, including the disseminated ones, limits the destruction strategy because the remaining cancer cells may redevelop the disease and even could build resistance to the applied chemo and radio methods. These challenges create a new approach, using the system’s defense and protective procedures against malignancy and mobilizing the body’s immune system through immuno-oncology.

#### 2.1.1. Change of Paradigm

We are in a war against cancer. The old military rule requests the attack of the enemy’s weakest point, but the conventional oncotherapies, including hyperthermia, attack the strongest malignancy side: proliferation. Change is necessary. Attack the weakest side: the missing networking, and consequently, the cancer is out from the overall regulation of the system. The cancer cells lost their collective connections and became autonomic.

Furthermore, the tumor-oriented curative approach has to be changed to the patient-oriented strategy to cure the patient, taking care of their complex health issues. Therapy must concentrate on human complexity and turn the product-oriented focus to the process-oriented one. This approach means that instead of concentrating on some molecular products, such as heat shock proteins, angiogenesis blockers, proliferation blockers, ionic blockers, blood-flow blockers, cell-poisoning, etc., we must focus on the dynamism of the cancer evolution process considering the immune effects, the physiologic feedback, and the overall homeostatic surveillance.

#### 2.1.2. Clinical Challenges

Unlike conventional hyperthermia, the mEHT treatment is recommended for:
Patients who cannot receive either surgery, chemo, or radiotherapy (conventional gold standards) according to various contraindicated aspects such as:
They have a comorbidity that contraindicates the conventional oncotherapy procedures.There is no effective conventional procedure for the given tumor.
Conventional curative procedures are no longer available and usually get a palliative setting only.
Patients with relapsed locally far-advanced tumors and no alternative standard curative therapy exists.Conventional therapy cannot be continued due to organ failure or low blood count.Despite standard treatments, patients show intense progression, relapse, and broad malignant dissemination.
Severe metastatic activity does not allow conventional treatment, salvage, or terminal state.

Due to the above challenges, most of the clinical trials conducted with mEHT are single-arm prospective or retrospective observational studies, meaning that it does not differ from other studies of medical devices [42].

The frequently applied single-arm studies have ethical and statistical reasons. Patients in this stage have no other treatments possible, and a cohort reference frequently does not exist for complete study statistics.

#### 2.1.3. Study Challenges

The evidence from single-arm observational studies is usually less convincing than that from randomized double-arm studies due to the huge variation of personal situations for patients with severe advanced stages of the disease. The personalized protocol cannot be rigidly fixed in preliminary planning. Having personal variations of the protocol makes the single-arm prospective trial difficult because the patient cohort and its protocol are not homogeneous. In these cases, only an observational study is available to indicate the efficacy of the treatment with an intention-to-treat (ITT) schedule on a carefully chosen personal basis. Despite the diverse cohort, the principal self-similarity allows the self-organized approach [43] which fits the allometric scaling by the fractal structure of the tumor [44]. In consequence, self-organization data mining could prove that the results have the necessary information to measure and evaluate survival [45]. Medical evaluations of survival conventionally apply Kaplan–Meier (KM) non-parametric estimator [46] for incomplete observations. KM is useful to examine the probability of lifetime and effectivity of the chosen treatment for such lethal diseases such as cancer. Taking the self-similarity into consideration, the hazard function must be a self-similar time function [47] and in consequence, the KM could be approached with Weibull distribution [48]

The invariance of magnification (scale invariance, when the up- or down-magnification show similar structures) is the form of self-similarity, which is a typical consequence of the self-organizing processes [49].

No “average” patient exists; the cohort is widely mixed. While in randomized studies, the randomization enables unbiased estimation of treatment effects; observational studies are typically not random. Propensity score matching (PSM) is a method of statistical analysis to estimate the effect of a treatment by accounting for the covariates that predict receiving the treatment. PSM is a conditional probability of being exposed given a set of covariates attempts to reduce the bias by the confounding variables [50]. The PSM improves the evidence level of the observation study, intending to reduce the treatment assignment bias by matching and mimicking randomization, by samples receiving the treatment that is comparable on all observed covariates without receiving the treatment. The possible reference solutions apply proper historical control from the same clinic/hospital where the observational study is performed, retrospectively choosing the same conditions. In the case of mEHT, the PSM was chosen from the patients from the same hospital with the same diseases and stages.

The PSM increases the trustworthiness of the obtained results [50,51] by combining them with an available database, selecting similar cases to be used as a control [52]. Selecting a comparative group of patients uses data mining in large and representative databases, defining the disease’s relevant and characteristic properties and the patients’ conditions.

The expectation that selecting the independent parameters from the actual therapy does not change during the complete curative or palliative process drives the propensity score comparison. The propensity score method gives statistical proof if the confounding variables are chosen well [53]. Advanced cancerous cases may limit the selection because of the large variety of previously failed treatments, so the applied database has to be large enough to mine the appropriate data.

Mathematical/statistical estimates may increase the single-arm’s strength of evidence. The single-arm has complete information about the patients [45,54], but their evaluation is difficult due to the missing comparison cohort. The self-organizing behavior of tumors provides a satisfactory accuracy of evaluating the single-arm, statistically deducting a reference group from the measured data [55].

Repeating the single-arm trial in different research places at different times for the same stage of the disease provides more realistic confidence supporting the evidence. The data pool of the different single-arm studies may increase the evidence level significantly. The most convincing statistical result is the similar, statistically equivalent survival curves of the studies performed at various times in various clinics and countries.

### 2.2. Clinical Results

The definite primary endpoint of the mEHT studies is the synergy of overall survival (OS) time with quality of life (QoL). Secondary endpoints are the local effects (response, remission, and local control). While the secondary endpoints are popular, they do not provide enough information about the patient’s overall status.

#### 2.2.1. Safety

The Phase 1 clinical trial safety measure was made with patients having advanced glioblastoma multiforme (GBM). This safety study approved the applicability of mEHT to such sensitive organs as the brain without remarkable side effects, even with drastic transcranial dose escalation [56], Table 1. Alkylating chemotherapy (ACNU, nimustin) was administered at a dose of $90\;mg/m^{2}$ on day 1 of 42 days for up to 6 cycles or until tumor progression Additional adverse effects of mEHT were not observed.

The GBM is a very fast-growing tumor that spreads rapidly to nearby normal brain tissue, but rarely forms extraneuronal metastases [57], so the distant dissemination to other organs is practically excluded. The selective behavior of the mEHT, and so the strict locality of the heating process, concentrates the energy on the malignant regions and the healthy tissues are unlikely to have harmful doses. Due to the nanoscopic heating, the common problem of the localization of the heat effect [58] is automatically controlled. The adverse effects were measured by dose escalation of mEHT complementary to the ACNU chemotherapy in four groups, increasing the weekly treatment dose from standard two/week to five/week.

The results showed convincing safety of the method, even with extra-large (practically not applied) doses [56]. An essential consequence of this safety trial is when a safe treatment of such a sensitive organ as the brain with such advanced disease as GBM could be performed, we may also expect safety for various other organs.

The radiotherapy combined mEHT trial (*n* = 20) was safe; no edema appeared with good local control of advanced GBM WHO Grade III–IV.

The optimal dose was determined by dose escalation of mEHT in the Phase I clinical study (*n* = 19) for patients with relapsed, refractory, or progressive heavily pretreated ovarian cancer. The dose optimum in this disease was 150 W/60 min [59].

Dose escalation of intravenous vitamin C (ivC) together with mEHT was measured with a Phase I safety study, which showed that 1.5 mg/kg ivC is safe for in Stage III–IV non-small cell lung cancer (NSCLC) patients [60].

A metabolically controlled complex therapy package of treatments could be effective in most advanced metastatic cases [61,62,63,64]. The monotherapy application of mEHT also presented promising results for patients with advanced disease when other therapies had failed [65,66].

#### 2.2.2. Survival Time

The clinical results of overall survival (OS) with the synergy of the quality of life (QoL) show the feasibility of complementary applications of mEHT with all standard adjuvant and neoadjuvant oncotherapies, including immuno-oncologic and integrative therapies.

The cervix cancers were studied in a Phase III double-arm randomized prospective controlled trial involving two–two groups in both arms, patients ±HIV [67,68]. Metabolic response (MR) was measured by PET [69]. The advantages of mEHT appeared in the higher complete metabolic response (CMR) of the HIV groups compared to the control groups. The continuation of a Phase III randomized controlled study for advanced cervical cancer patients provides new insights through the strong evidence of mEHT’s efficacy to improve the three-year overall survival [70], Table 2. The OS for patients with FIGO Stage III disease had a significant (*p* = 0.04) increase with mEHT addition compared to without it. The disease-free survival (DFS) was also significantly longer (*p* = 0.04) for these patients.

A Phase II randomized double-arm study compared platinum-based chemotherapy to additional complementary mEHT for patients with recurrent cervical cancer, dominantly disseminated squamous cell carcinoma, in a broad range of FIGO statuses [71]. The obtained overall remission rate at seven months of follow-up was significantly better in the active mEHT arm (*p* = 0.02), and despite of great difference, the overall survival had not reached the significance level (*p* = 0.24) [72], probably due to the small number (n = 20 + 18) of participants). FIGO Stage 2 locally advanced cervical cancer with lymph node metastases in more than half of the treated patients was studied in retrospective observational studies in a double-arm comparison of standard chemoradiotherapy and its extension with mEHT [73]. The results are convincing (Table 3).

A comparison of the mEHT-treated and nontreated patients with the same stage brain tumors in the same research group in Italy [74] showed a highly significant (*p* = 0.0006) response rate compared to the conventional control and also for GBM (*p* = 0.026). The details of the study are shown in Table 4. The theoretical self-similar reference arm corresponds well with the KM plot of the nontreated group of patients [55].

Retrospective observational study with mEHT-treated and untreated arms for advanced pancreatic cancer showed that mEHT was successfully applicable for various pancreatic tumors [75], and also for the specially nonresectable tumors compared to PSM [76], as well as relative long overall survival; Table 5.

A randomized double-arm Phase II study (*n* = 49 + 48) showed significant improvement in survival and quality of life for patients with Stage III-IV NSCLC treated with mEHT in combination with high-dose vitamin C infusion and providing the best supportive care (BSC) in both arms [79]. The three months follow-up remission rate was significantly better (*p* = 0.0073) in the active mEHT arm than without it, and simultaneously the survival was also significantly (*p* < 0.0001) improved. A randomized two arms Phase II clinical trial for advanced non-small-cell lung cancer (NSCLC) patients also showed a clear advantage of mEHT application, showing a significant increase (*p* < 0.001) in the patients’ OS [79].

The mEHT method also demonstrates the feasibility of treating advanced small-cell lung cancer (SCLC) patients [80], where a significant (*p* = 0.02) increase in overall survival was observed compared to the control arm of the study.

The mEHT can also be used advantageously in gastrointestinal cases [62,64,81]. The first-line, single-arm, retrospective clinical study (*n* = 40) of metastatic colon cancer complementary to Bevacizumab+FOLFOX [82] observed progression-free survival (PFS) of 12.1 months and OS 21.4 months, which are remarkably good results.

Neoadjuvant (preoperative) mEHT treatment for locally advanced rectal cancer was studied in a double-arm trial (*n* = 62 + 58) [83]. The tumor regression was significantly (*p* = 0.0086) decreased by the tumor volume in the control arm, while in active mEHT treatments, the regression grade was uniform (*p* = 0.91) and independent of the tumor size. Despite lower radiation doses in the mEHT group, the clinical measures were comparable to the control group; the proportion of downstaging (80.7% vs. 67.2%) and pathologically complete response (pCR, not only for imaging) (17.7% vs. 8.6%) was higher with mEHT than without it. The pathological T-stage (ypT) was significantly (*p* = 0.049) better with mEHT, and also the rejection margin was significantly (*p* = 0.013) improved by mEHT application. The survival measures (overall, disease-free, local recurrence-free, metastatic recurrence-free) were all improved by the mEHT, but the differences did not reach a significant level (*p* = 0.05).

Another Phase II single-arm (*n* = 60) clinical trial for neoadjuvant mEHT was performed for rectal cancer in cT3-4 or cT2N+ stages [77]. The therapy showed T- and N-downstaging in 40 patients (66.7%) and 53 patients (88.3%), respectively. In total, 15% of patients had complete pathologic response in the T-stage, and 76.7% in the N-stage.

The treatment of peritoneal carcinomatosis with malignant ascites with mEHT combined with traditional Chinese medicine compared to intraperitoneal chemoinfusion (IPCI) in a Phase II randomized double-arm trial [78] observed a better overall response (77.7% in the study arm, while 63.8% in control).

Notably, for cholangiocarcinoma [84,85] and tumors of the hepatopancreatobiliary system [38], treatments with mEHT show the feasibility of the method on presently low-success curing-rate tumor localizations. Two Phase II studies proved the successful applicability of mEHT in hepatocellular carcinoma [86,87].

The mEHT was successfully applied to advanced breast cancer [88], including triple-negative cases [89,90], and leiomyosarcoma of the breast [91]. The study of thirteen patients with complicated, advanced invasive ductal breast carcinoma, mostly triple-negative immunohistochemical status and multiple metastases, showed more than two months of median survival [92].

A study of advanced ovarian cancer treated with mEHT complementary combined with paclitaxel and cisplatin chemotherapy showed less toxicity and adverse effects than cisplatin [93]. The combined two drugs showed no Grade III or IV toxicity and a 57% remission rate during 30 months follow-up, together with 85.7% survival in the same period [94]. A Phase II study of heavily pretreated, mostly platinum-resistant ovarian cancer with relapsed, refractory, or progressive stages treated with mEHT, showed a remarkable 7.5 months median survival in the same treatment period [59].

#### 2.2.3. Comparison of Survival Curves

Glioblastoma treatments with mEHT have numerous single-arm prospective and retrospective clinical studies [74,95,96,97,98,99,100,101,102]. The studies had the same treatment protocol in Figure 2.

Multiple single-arm studies are appropriate for comparison. Evaluating the comparison of these results showed an excellent match with each other, Figure 3. The similarity of the curves may mean strong evidence of the mEHT success.

The various GBM studies following the same general protocol are comparable, Table 6.

A meta-analysis also showed the superiority of the mEHT treatment [104]. Furthermore, the results could be compared with the effective update chemotherapy of GBM, the temozolomide (TMZ) [105], Figure 4. The comparison with the pooled data shows the advantage of mEHT again.

Furthermore, the pool of survival rates of the GBM patients (*n* = 325) in various single-arm studies showed good agreement with the invasive transcranial brachytherapy ± invasive hyperthermia of the same disease [106], Figure 5. The invasive method (brachytherapy alone or combined with invasive hyperthermia (iHT)) do not differ from the survival results of noninvasive mEHT.

The tumor treating field (TTF) is an emerging electromagnetic therapy for GBM, showing the electric field’s efficacy for this disease [107]. Its focal point is the cytokinetic “neck” at the end of the mitotic spindle, applying nonthermal effects with capacitive coupling [108]. The electric field of TTF reorients the highly polarizable microtubules and actin fibers, and it may arrest the cytoskeleton’s polymerization process and inhibit the assembling of the mitotic spindle [109]. Impressive clinical results were achieved with TTF, proving the feasibility of the nonthermal application of bioelectromagnetic processes against malignant proliferation. A comparative meta-analysis of TTF and mEHT [110] establish that the use of both mEHT and TTF in the treatment of glioblastomas can improve overall survival. A comparison of the mEHT pool and TTF results [111] showed no difference in the clinical results of the two electromagnetic therapies, Figure 6. The differences between TTF and mEHT are basically in the practical application and length to complete the therapy. TTF fixes a hat with multiple electrodes which the patient has to wear 18 h/day for several months, while mEHT makes intensive 60 min treatments every second days in 12 sessions.

The successes of the complex therapy involving mEHT appear as some challenges for randomized studies of GMB [112]. An intensive naturopathic-oncotherapy (Bevacizumab + Boswellia serrata + Curcumin) combination with mEHT looks feasible for GBM patients in a terminal state [113]. This was measured earlier also using high-dose IV Vitamin C (30 g) combined with Thalidomide (50 mg) and Boswellia serrata (400 mg) plus fortecortin (0.5 mg) [114] in cases for patients for whom the chemotherapy is contraindicated.

The GBM has only rarely metastases regardless of the ability of GBM cells to be disseminated over the blood–brain barrier (BBB) [115]. This specialty makes the disease mostly localized, and the various metastases only rarely influence the survival time, which is otherwise the drastic limiting factor of survival in other malignancies. Due to these conditions, we do not expect such unified survival probabilities in other tumors as we observed in GBM.

There are protocols and guidelines not only for the therapy of central nervous system, but those available for various other applied therapies as well [116,117].

We compared the various pancreas studies collected in Table 7. The treatment success of advanced pancreatic cancer has shown relatively little development recently. The applied protocols are that most pancreatic malignancies are non-resectable or can only be partially excised (R1) and frequently make liver metastases, increasing the mortality of this disease. Intensive clinical research with mEHT is underway in this area [61,118,119,120,121]. A comparison of the Kaplan–Meier survival curves shows a definite similarity in the survival probabilities, Figure 7, without such unified curves as we observed in GBM.

Another massively present malignancy in the world is advanced non-small-cell lung cancer (NSCLC). Application of mEHT on advanced NSCLCs also has numerous case studies [63,122,123,124,125,126] and trials [60,79,127,128,129,130]. The comparison of the survival distributions shows similarities again, and the data pool remarkably differs from the propensity score data by mining in USA and EU databases [130], Figure 8.

The comparison of the survival plots from different NSCLC studies Figure 9, Table 8, is not as unified as the GBM comparison in Figure 1, because of the multiple variations of metastases and tumor locations in the lung. However, no significant differences appeared in the observed survival probabilities, so the pooling of data was also possible.

The liver metastases of colorectal carcinoma appear to be a frequent complication in this malignancy. Multiple trials have demonstrated the efficacy of mEHT in this case, too [99,132,133]. The colorectal carcinoma survival studies also may be compared Table 9.

The comparison of the measured Kaplan–Meier non-parametric distributions from different studies, Figure 10, had no significant differences but differed considerably due to the high variability of these metastatic conditions. The pooling here had no statistical basis.

#### 2.2.4. Quality of Life

The synergy of OS with QoL is especially important in the less-successful conventional therapies, such as the brain, pancreas, lung, and liver, which are otherwise frequent metastatic locations from various malignancies when their mortality is exceptionally high.

The Phase I safety study (*n* = 35, involving Stage IV *n* = 17) for NSCLC observed adverse effects (fatigue, nausea, vomiting, diarrhea, headache) only rarely and also temporarily [60]. The function subscale of QLQ-C30 scores showed significant improvement in physical status after four weeks of treatments compared to before the therapy and getting gains in all other categories (emotional, cognitive, social, global) without reaching the *p* < 0.05 significance level. However, the advantage results of mEHT on the symptoms subscale were significant after four weeks in most categories (such as fatigue, dyspnea, insomnia, appetite, and diarrhea). The decrease in nausea/vomiting, pain, and constipation were also observed without significance. The Phase II continuation of the same NSCLC study [79] compared the QoL data in time, not to the baseline. The control arm for comparison was received by randomization of the included cohort of patients. In this randomized study, the QoL significantly improved in all conditional (functional) categories and physiometric (symptoms) categories.

The data in the comparison of IPCI with mEHT favored the latter by improved overall QoL with 32.3% and 49.2%, respectively [78]. Primarily, pain relief in lung cancer was studied in a propensity case-controlled study [124], showing increasing pain after mEHT, which gradually lowered, and after a long time, the effective analgesic score decreased from the original 8.5 to −83.7%, which was more than a 90% improvement compared to baseline before therapy.

Similar results were obtained with less radiation dose combined with mEHT than alone [77,83], which may result in fewer adverse effects and higher QoL of the treated patients.

Improved QoL functions were measured with a Phase III randomized prospective, controlled clinical study of the same cohort of patients showing decreasing toxicity and increased quality of life [134]. Six weeks after the mEHT therapy, the cognitive function significantly improved compared to the control arm. A significant increase in social and emotional function was measured three months post-treatment, while fatigue and pain were significantly decreased simultaneously. Notable the QoL function was also improved and the ratio of mEHT-arm to the control grew by 1.6, 1.3, 2, 2, 10.8, 1.7, and 5.7 for visual analog, global health, pain reduction, nausea/vomiting reduction, fatigue reduction, cognitive functions, emotional functions, role function, and physical status, respectively.

#### 2.2.5. Immunogenic Effects

The immunogenic effects of hyperthermia are currently at the forefront of research and fit well into the general trend that immunology would require. However, hyperthermia applied by itself can also produce immunogenic effects. Cancer patients usually have a weakened immune system that requires stimulation. One of the first series of hyperthermia boosted by immunostimulants was published in 1986 [135]. The double-arm study (*n* = 77, 75%, surgery, 25% inoperable) of pancreas adenocarcinoma applied capacitive coupling hyperthermia 13.56 MHz plus chemotherapy (doxorubicin 30 mg/m2, 5-FU 500 mg/m^2^, mitomycin C 5 mg/m^2^). In total, 42% of patients had chemotherapy in the inoperable group, 5% radiotherapy, 21% radio-chemotherapy, and 32% had no prior therapy due to their refractive status. The immune stimulation used granulocyte–macrophage colony-stimulating factor (GM-CSF) and the results were very encouraging. The ratios of survival percentages between immune-treated and untreated groups showed 2.0-, 5.8-, and 3.0-times increase of 6, 12, and 18 months survival by immune boosting, respectively. The results were highly significant.

The primary task of immunogenic stimulation is to restore homeostatic control and ensure the systemic surveillance of healthy processes. The local treatment becomes systematic in this way and allows attacking the distant micro- and macro-metastases, forming abscopal outcomes [35]. The abscopal principle could be used as a new anti-cancer vaccination strategy with immune stimuli [136], emerging “hyperthermic immunotherapy” [137], and developing tumor-directed immunotherapy [138]. Studies have shown that combining mEHT with traditional Chinese medicine as an immune booster also has abscopal effects [139].

When the patients’ immune system is strong enough to develop tumor-specific immune reactions, then the mEHT works without extra immune stimulation. A Phase III clinical trial for uterus cervix cancer proved the abscopal phenomenon [68,140], forming the complete metabolic response almost five-times more than the otherwise systemic chemotherapy in the control arm. The extra-pelvic response abscopal effect does not depend on the HIV status of the patient. The mEHT treatment of colon cancer clearly shows an abscopal effect in liver metastases [82].

A new therapeutic field is the complementarity application of checkpoint inhibitors (CPIs) combined with mEHT [141,142]. The immune action by CPI leads to a definite abscopal effect in clinical practice [143,144].

Viral immunostimulant with Newcastle viruses is a new, emerging complementary treatment with mEHT, allowing a new strategy for whole-body action [145]. The mEHT has a complex synergy with viral immune stimulation [146,147], also using the ICD process for developing tumor-specific immune activity [148]. The mEHT is a part of multimodal immunotherapy for patients with GBM [149], allowing personalized medicine in glioblastoma multiforme [150]. The complex therapy by dendritic cell vaccines and other immune stimulation to develop ICD within chemotherapy administration might improve the overall survival rate of GBM patients with long-term tumor control [112,151], as well as the induction of ICD during maintenance chemotherapy combined with subsequent multimodal immunotherapy for GBM. A study (*n* = 41) showed the remarkable benefits of mEHT as part of multimodal immunotherapy for brain tumors in children with DIPG [152], without significant toxicity. The median PFS and OS were 8.4 m and 14.4 m from the time of diagnosis, respectively. The two-year OS was 10.7%. Immunotherapy was applied at the time of progression, when the measured PFS and OS medians were 6.5 m and 9.1 m, respectively.

## 3. Discussion

The evidence level is improved by comparing the survival data of various studies from different institutes. The relationship between relevant higher evidence-level studies and large databases gives efficacy information. The data of the variant of single-arm observational studies were pooled and used to compare to another type of treatment results. The pooling of data is correct because these survival plots do not vary significantly. The pool of data mimicked the later introduced market surveillance when the various results from very different medical groups were compared. The well-correlating single-arm survival showed that the application of the same protocol at different hospitals, countries, and time provided statistically equivalent data, so it is a usual requirement for worldwide approved drug applications. The technical solution allows easy and safe application [153].

The comparison of Kaplan–Meier survival times of glioblastoma multiforme in six studies made in different institutes gave an excellent agreement between the curves. The match was probably so accurate because the GBM rarely gives non-neural metastases [57], so the distant metastases do not modify the survival. However, the micro- and macro-metastases could critically worsen the survival time. The frequent distant metastases of these tumors are the reason why we do not have perfect equivalence to the KM survival plot in pancreas carcinoma, NSCLC, and colorectal carcinoma, although the differences remain insignificant. The practically identical survival results increase the evidence level of the studies. Due to the insignificant differences, pooled data may be used for comparison with other kinds of cancer treatments for the same tumors. The pooled data showed the superiority of mEHT over temozolomide + radiotherapy treatment for GBM. The mEHT pool agreed well with invasive hyperthermia and the results of other nonthermal bioelectromagnetic therapy (TTF).

The immune stimuli and the immunogenic cancer cell-death [154] are probably a considerable addition to the elongation of the patient’s survival.

The quality-of-life measurements had not revealed extra adverse effects of mEHT and even showed improvements in all functional and physiometric symptomatic scores. The preclinically approved immunogenic effects of mEHT [136,155,156,157], the human response studies [137,141,142], and the observed abscopal effects [138,143,144] probably promote the presented improvement of survival and QoL.

The mEHT, as the synergic therapy of thermal and nonthermal effects, such as other therapies, has limitations. Various challenges appear in general hyperthermia treatments [158,159], which exist and are combined with some specialties in mEHT. The average power of mEHT has to be limited for optimal treatment and the energy absorption has to be balanced, due to:As natural nanoparticles in the membrane, the rafts are heat-sensitive molecular clusters. The too-large absorbed energy destroys the rafts by overheating. The massive distortion of the rafts may degrade the membrane integrity and cause necrosis, losing the apoptotic “harmony” with the homeostasis, which is suboptimal.The selected energy absorption of rafts heats the TME and tissues to a lesser extent. The standard applied SAR in nanoparticles, considering their weight heating, is 0.1–1.5 MW/kg. The approximation of the absorbed power of rafts in selection is SAR > 1 MW/kg [19], which is similar to the standard MNP energies [160]. The increased diffusion redistributes the initial spacing with nanoparticles [161]. The electric field impacts the diffusion of the charged and dipole particles, modifying the electrokinetics of the effusion [162] and the angiogenesis [163]. In case of electric field heating at mEHT, the electrodiffusion modifies the allocation of the gold NP-s too, positions them to the volumes of high electric field, thus promoting the heat on the TME [162]. The heating of the NPs shares the energy, reducing the effect on the membrane rafts, and, despite the increase of temperature, the apoptosis decreases [164]. The distribution of magnetic NP-s, which are modified by the increased diffusion with the temperature [165], has to be impacted too by electrophoresis and electroosmosis, and so the electric field in low frequencies (modulation frequencies of mEHT) regroups them on the same way such as in the case of the non-magnetic metallic NPs [164].The thermal effect happens in nanoscopic local “points”, the rafts. These molecular clusters are sensitive to overheating. When the absorbed energy is too large, it destroys the rafts and the mEHT loses its largest advantage, producing immunogenic cell death (ICD).The large energy absorption extensively forces the spread of heat, and the selection of microscopic differences vanishes. A macroscopic average will characterize the target, and the cellular selection with intended molecular excitations will vanish. The thermal component will become dominant, and the selection mechanisms cannot prevail. A limited thermal component ensures the selection of rafts.

The thermal conditions induce numerous physiological processes interacting with the body’s thermal regulation, which limits the temperature gain (∆T) with the following processes:For human adults the surface heat-loss is f_loss_≅0.15 at rest in the 0 ≤ f_loss_ ≤ 1 scale [166], so the heat exchange is intensive enough even by intense local heating. Consequently, the bloodstream in the tumor maintains massive cooling efficacy. The cooling is inhomogeneous; it depends on the vascularization. As such, the thermal factor of mEHT is less homogenous than the nonthermal one.The thermally induced vasoconstriction regulates the blood perfusion and heat conduction in tumors [167,168,169], while the heated healthy tissues in the surroundings have vasodilation. The relative blood flow could promote vascular invasion of the tumor border, reducing the prognostic expectations [170]. The special nano-selectivity and the applied low incident power (about 1/8th of the other conventional local hyperthermia therapies) produce a moderate (fever level) temperature in the tumor mass and its surrounding tissues, which causes much less increase of the blood perfusion than the conventional hyperthermia methods, Figure 11. The thermal damage, which is usually calculated [171], is not considered in mEHT, the nonthermal factor makes the apoptosis, ICD, and immune effects also have a gradient on the tumor border, but it helps focus on the denser tumor by refraction angle.The thermally promoted intensive metabolic activity deprives the ATP sources [172]. However, the massive energy demand of proliferation requests enormous ATP production, which induces anaerobic metabolism, improving the intensive proliferation [173], promoting the malignant processes [174], and leading the growth direction by an acidic invasion front [175].A positive process of ATP deprivation may cause protein aggregation in the cytosol [129,176], destroying the cytoskeleton order. The collapsing cytoskeleton destabilizes the plasma membrane, and the cell necrotizes [176]. Increased temperatures can slow down or even block DNA replication [177], and the DNA strand breaks [178], which completes radiotherapy [179].The intensive thermal effect acts mainly in the S-phase of the cell division cycle [180], while moderate heat shock arrests G1/S and G2/M cell-cycle checkpoints [181]. The nonthermal factor has an intensive block of the last phase of the cell cycle.

Furthermore, the mEHT application must consider the following conditions to avoid the suboptimal treatment:The appropriate frequency is accurately selected around 10 MHz [182,183]. When the frequency is larger (>15 MHz), the membrane impedance becomes too small to select the disordered TME. The current will flow through the entire cell almost equally, neglecting the selection factor of dielectric permittivity.The electromagnetic nano-targeting of rafts has similarities to the molecular targeting of drugs at cancer cells. The chemo dose is limited by poisoning. When the rafts are overheated, the raft protein may coagulate, and no selective heating is possible thereafter.Hyperthermia may reversibly destabilize the raft structures [184], which could mix the time sequences of DAMP production and arrest the immune cells’ activity [185].The approaching of the contact current by mEHT has further limitations. RF safety standards specify the exposure limits [186]. The SAR could be extremely high in the small cross-section when the applicator does not smoothly cover the treatment area and the current flows through a small area which may burn in this touching. The challenge grows when the interface between the skin and electrode has a conductive layer such as sweat, saline, or other aqueous solution. The thin layer may be heated dangerously quickly, so the skin surface must be kept dry.

## 4. Conclusions

The success of the mEHT therapy of advanced, relapsed, and metastatic tumors has had multiple clinical studies. Some studies have double-arm comparisons, including a Phase III randomized, prospective controlled one for the uterus and cervix. The comparisons show a significant increase in survival time. The Kaplan–Meier survival plots were compared for the single-arm studies. The different studies at various institutes’ and times showed significant correspondence, and the data could be pooled, which increases the evidence level of these observational studies. These clearly show the survival advantage of mEHT. We may conclude that mEHT significantly increases overall survival and the quality of life; consequently, it is a feasible treatment for the presented malignant tumors.

## Figures and Tables

**Figure 1 cancers-15-04569-f001:**
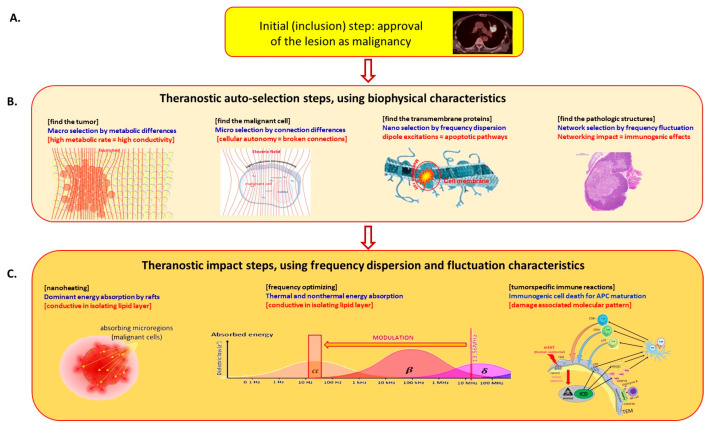
Selective and theranostic behaviour of mEHT. (**A**) The malignancy is localized and proven with conventional methods. (**B**) The RF current macroscopically selects the tumor by electric conductivity and microscopically by dielectric permittivity. The transmembrane proteins thermally and nonthermally absorb the energy in the malignant cells. The pathologic irregularities further increase the selection by applying RF current. (**C**) The nanoscopic membrane rafts locally heat the malignant cells, and the optimized modulated RF current makes the desired molecular changes for DAMP and ICD. The DAMP molecules induce antigen presentation and, as a consequence, antitumoral killer and helper T-cells appear, working as a tumor-specific vaccination.

**Figure 2 cancers-15-04569-f002:**
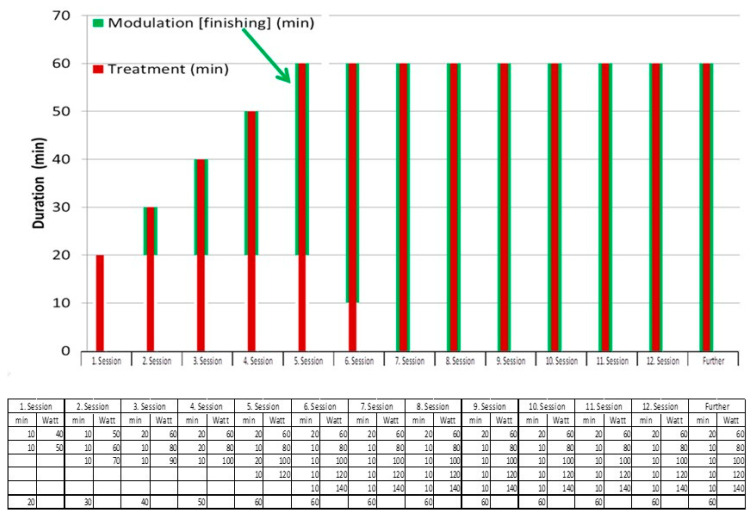
The protocol for the GBM treatment. The modulation of the carrier frequency, and the energy load had to be adapted by the patient (green columns).

**Figure 3 cancers-15-04569-f003:**
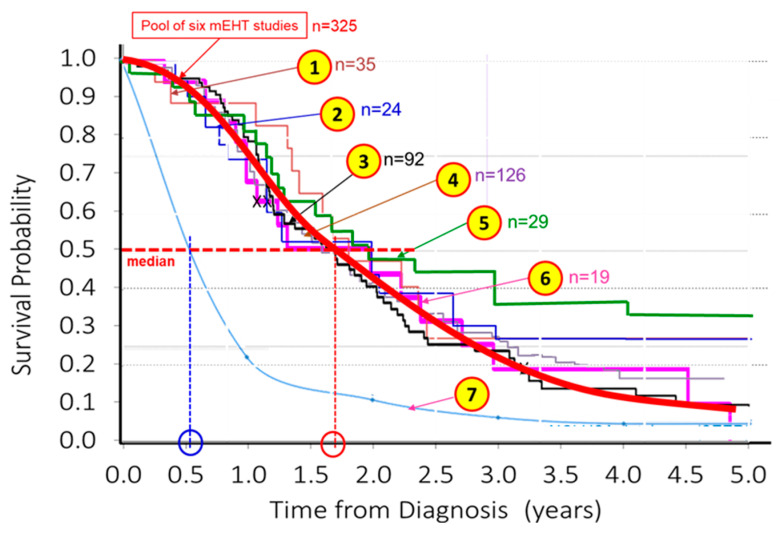
The survival results at different times and various clinics. The Kaplan–Meier probability comparison showed no statistical difference between the different clinical studies, so the data may be pooled containing 325 patients altogether. The average median value was significantly higher than the SEER database (data from [61,99]). (① = [99], ② = [96] ③ = [95] ④ = [101] ⑤ = [103], ⑥ = [97], ⑦ = SEER (NCI, USA) data from [99].).

**Figure 4 cancers-15-04569-f004:**
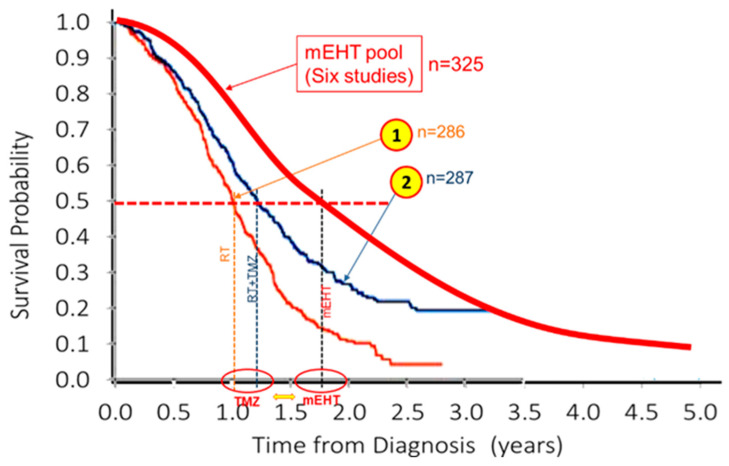
Comparison of mEHT pooled GBM survival probability with the literature [105]. ① is the reference arm (radiotherapy (RT) alone) and ② is the active arm, RT + TMZ.

**Figure 5 cancers-15-04569-f005:**
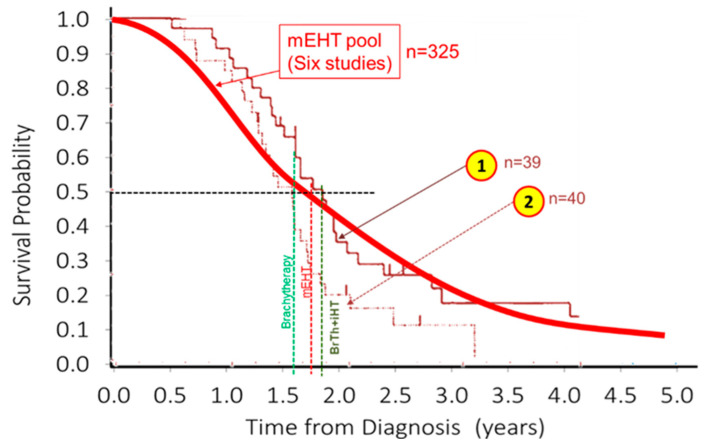
The invasive brachytherapy and invasive hyperthermia (iHT) in brain for GBM [106] do not differ from the mEHT pooled data. ① = brachytherapy (bTh) alone and ② = bTh + iHT.

**Figure 6 cancers-15-04569-f006:**
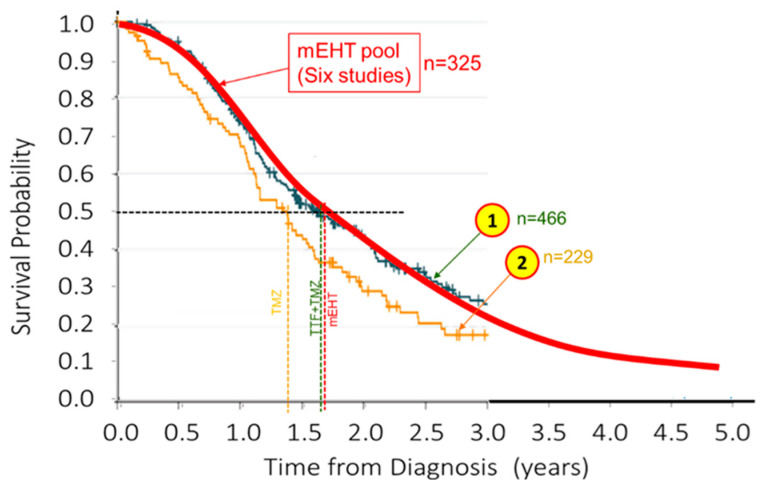
Comparison of the pooled data of six mEHT studies to TTF+TMZ survival data [72,111], where ① shows the result of the active study arm TTF+TMZ and ② is the reference with TMZ alone.

**Figure 7 cancers-15-04569-f007:**
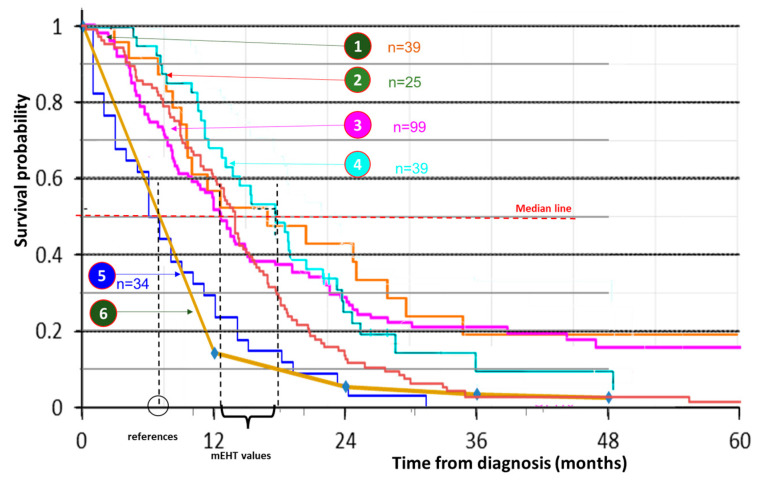
Comparison of the survival studies of advanced pancreas carcinomas ① [75] (*n* = 106), ② [99] (*n* = 25), ③ [120] (*n* = 99), ④ [76] (*n* = 78), ⑤ [120] (*n* = 34), ⑥ SEER (NCI, USA) data from [99].

**Figure 8 cancers-15-04569-f008:**
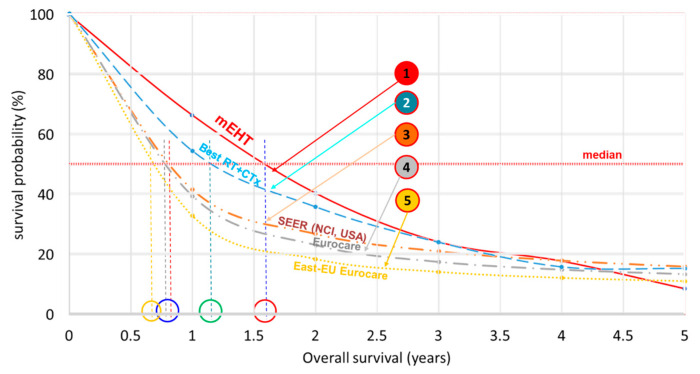
Comparison of the NSCLC survivals from large databases [92,130]. ① mEHT pool, ② Best RT+ CTx, ③ SEER (NCI, USA), ④ Eurocare-5, ⑤ East-EU, Eurocare-5.

**Figure 9 cancers-15-04569-f009:**
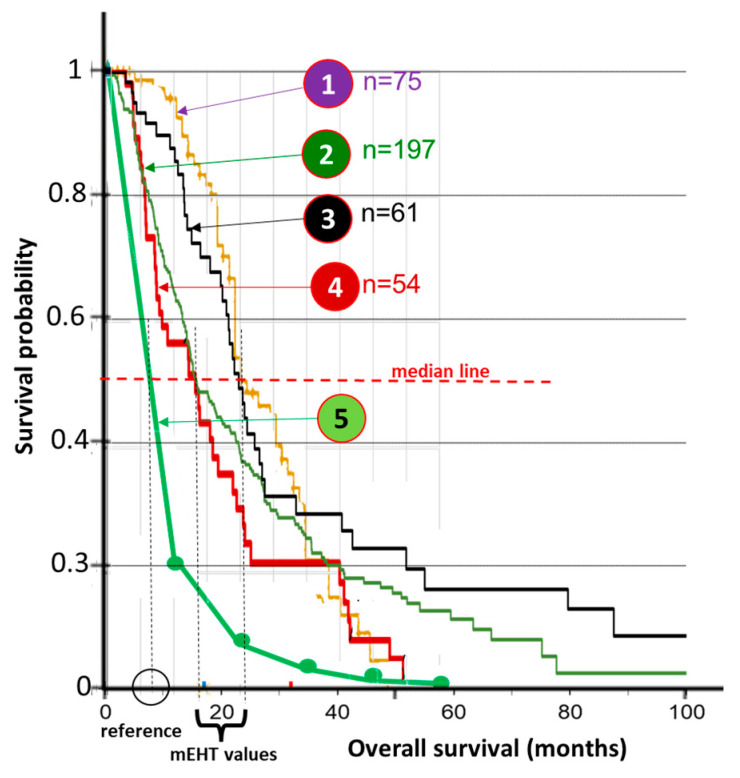
The measured NSCLC survival probabilities. ① = [127], ② = [129], ③ = [131], ④ = [61,99], ⑤ = SEER (NCI, USA) data from [99].

**Figure 10 cancers-15-04569-f010:**
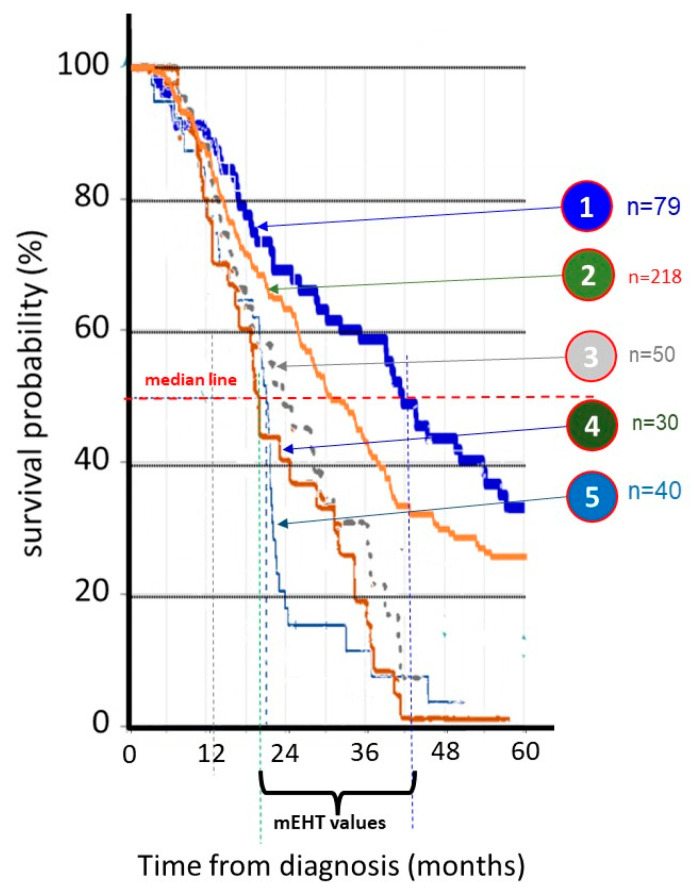
The colorectal cancer survival plots. ① = [99], ② = [133], ③ = [132], ④ = [132], ⑤ = [82].

**Figure 11 cancers-15-04569-f011:**
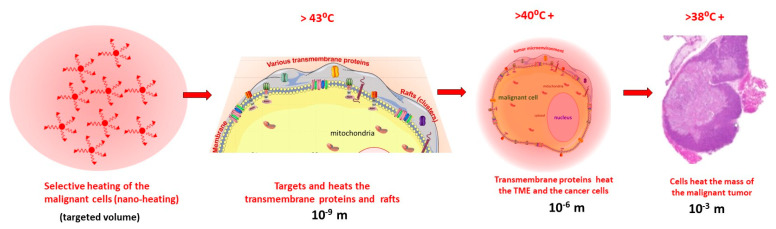
Heating heterogeneity. The characteristic sizes and the expected temperatures are shown in the various steps.

**Table 1 cancers-15-04569-t001:** The dose escalation of the GBM treatment with mEHT. Advanced GBM 3rd and 4th line treatment was studied. The complementary chemotherapy was nitrosourea drug Nimustine (ACNU) $90\;mg/m^{2}$. Patients were grouped in 4 disjunct study arms by dosing for 6 weeks cycles.

Group	Number of Patients	mEHT/Week (6 Cycles)
1	4	2
2	4	3
3	4	4
4	3	5

**Table 2 cancers-15-04569-t002:** Survival results of the Phase III uterus/cervix cancer study [67,68].

Groups		Number of Patients	%	Average Age (y)	3 y Overall Survival (%)	*p*-Value
All		210	100			
RT + ChT alone	HIV positive	55	52.9	50.6	33.7	0.04
	HiV negative	49	47.1			
RT + ChT + mEHT	HIV positive	52	49.1	49.2	44	
	HiV negative	54	50.9			

**Table 3 cancers-15-04569-t003:** Results of retrospective double-arm observational study for cervix carcinoma. (CR-complete remission, DFS disease-free survival, NED—no evidence of disease.

Groups	Number of Patients	%	OS after 5 y		CR with Lymph Node Mets.		CR (NED)		DFS after 5 y		DFS with Lymph Node Mets. after 5 y	
All	95	100	%	*p*-value	%	*p*-value	%	*p*-value	%	*p*-value	%	*p*-value
RT + ChT alone	50	53	79.5	0.079	45	0.0377	58	0.0315	73	0.166	73	0.166
RT + ChT + mEHT	40	42	81		71		82		80		80	

**Table 4 cancers-15-04569-t004:** The response data of the treatments. AST—astrocytoma (Grade III), CR—complete remission, PR—partial remission, SD—stable disease, PD—progressive disease, OS—overall survival.

Response	Astrocytoma				Glioblastoma			
	without mEHT (*n*, %)		with mEHT (*n*, %)		without mEHT (*n*, %)		with mEHT (*n*, %)	
	*n*	%	*n*	%	*n*	%	*n*	%
CR	6	28.6	2	6.9	2	2.4	1	3.4
PR	1	4.8	10	34.5	2	2.4	6	20.7
SD	5	23.8	9	31.0	13	15.3	11	37.9
PD	8	38.1	6	20.7	63	74.1	11	37.9
No data	1	4.8	2	6.9	5	5.9	0	0.0
OS median (months)	17		72		12		15	
OS range	3–120		3–156		2–84		2–108	
*p*-value	0.0006				0.026			

**Table 5 cancers-15-04569-t005:** The main parameters of two survival studies [77,78] (*—macro metastases, ** possible micro metastases.).

Response	Fiorentini et al.				Petenyi et al.			
	without mEHT (*n*, %)		with mEHT (*n*, %)		without mEHT (*n*, %)		with mEHT (*n*, %)	
	*n*	%	*n*	%	*n*	%	*n*	%
Patients no.	67		39		39		39	
Males	38	56.7	24	61.5	19	46.2	18	46.2
Females	29	43.3	15	38.5	20	53.8	21	55.8
Age (mean, y)	66		61.8		66.02		65.9	
Distant metastasis *	37	55.2	25	64.1	24	61.5	20	51.3
Non metastatic **	30	44.8	14	35.9	15	38.5	19	48.7
Gemcitabine combination	64	95.5	27	69.2	31	79.5	31	79.5
Other complementary	3	4.5	12	30.8	8	20.5	8	20.5
OS median (months)	10.9		18		10.58		17.02	
OS range	0.4–55.4		1.5–68		2.4–48.8		4.4–47.1	
*p*	0.00165				0.0301			

**Table 6 cancers-15-04569-t006:** The GBM studies which were used for KM comparison in comparison Figure 1.

No.	Number of Patients	Treatments	OS Median (Months)	Reference
1	35	mEHT + RT + ChT + BST	26.4	Parmar, et al. 2020 [99]
2	28	mEHT + RT + ChT + BSC, (palliative)	14	Fiorentini, et al. 2018 [96]
3	92	mEHT + RT + ChT	20.4	Sahinbas, et al. 2007 [95]
4	126	mEHT + RT + ChT	20.3	Hager, et al. 2008 [101]
5	29	mEHT + RT + ChT	14	Szasz, et al. 2010 [103]
6	19	ChT (ACNU)	21.8	Douwes, et al. 2006 [97]

**Table 7 cancers-15-04569-t007:** The pancreas carcinoma studies which are used for KM comparison in comparison in Figure 5.

No.	Number of Patients	Treatments	OS Median (Months)	Reference
1	39	GMZ combination with mEHT	18	Fiorentini et al. 2019, [75]
2	27	GMZ combination with mEHT	13.2	Parmar et al. 2020 [99]
3	99	GMZ combination with mEHT	12	Dani et al. 2012 [120]
4	39	GMZ combination with mEHT	17	Petenyi et al. 2021 [76]
5	34	GMZ combination without mEHT	6.5	Dani et al. 2012 [120]

**Table 8 cancers-15-04569-t008:** The survival data of NSCLC studies.

No.	Number of Patients	Treatments	OS Median (Months)	Reference
1	75	RT + ChT + OP with mEHT	16.4	Szasz, 2014 [127]
2	197	RT + ChT + OP with mEHT	15.6	Dani, et al. 2012 [129]
3	61	RT + ChT + OP with mEHT	16.4	Dani, et al. 2009 [131]
4	54	RT + ChT + BSC with mEHT	18	Parmar, et al. 2020 [99]

**Table 9 cancers-15-04569-t009:** Survival data of colorectal studies.

No.	Number of Patients	Treatments	OS Median (Months)	Reference
1	79	ChT with mEHT	48	Parmar, et al. 2020 [99]
2	218	OP + ChT with mEHT	28.5	Szasz, et al. 2010 [133]
3	50	BSC with mEHT	25	Hager, et al. 2020 [132]
4	30	ChT + BSC with mEHT	23	Hager, et al. 2020 [132]
5	40	ChT with mEHT	21.4	Ranieri, et al. 2020 [82]

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
