# Peer review of "The Clinical Validation of Modulated Electro-Hyperthermia (mEHT)"

_cancers, 2023, doi:10.3390/cancers15184569_

Round 1

Reviewer 1 Report

1.       For a better and more comprehensive overview the authors should enlist all the studies that they are referring to in a table format with the major findings, references , year of the study, number of the cases etc.

2.       Based on the reviewer’s search, four trials using mEHT were registered in the ClinicalTrials.gov repository, which are also discussed in the manuscript. Authors should discuss this, use and refer to the Identifier of the study. If other trials were registered elsewhere, it should be mentioned as well.

Author Response

Thank you very much for your valuable comments. We had collected the studies in table grouping, and extended all the references. The submission has altogether a longer version with the extension.

Thanks again for your critical proposals.

On behalf of the authors:

Marcell A. Szasz

Reviewer 2 Report

Peer Review Report

Manuscript ID: Cancers-2452603

Title: The clinical validation of modulated electro-hyperthermia (mEHT)

The study by authors Lee et al. lies within the journal scope of Cancers. The authors present the clinical validation of modulated electro-hyperthermia (mEHT). The study summarizes phase III trial of mEHT for advanced cervix tumors of HIV-negative and positive patients. Furthermore, the efficacy of mEHT was explained for pancreatic, lung, colorectal and glioblastoma tumors. There are several suggestions for the authors to incorporate before recommending the work for publication with the journal of “Cancers”. In the first read, it is difficult to follow and have technical jargons which should be avoided for broader readership.

1. Add more context to the Introduction. Revise this section. It can be improved.

2. Compare and contrast single-arm observational and randomized double-arm studies?

In reference to (Lines 124-127), seems contradicting to each other. Explain?

Due to the huge variation of personal situations, the protocol cannot be rigidly fixed, which makes difficulties of the single-arm prospective trial for patients with such severe states. In these cases, only an observational study could show the efficacy of the treatment with an intention-to-treat (ITT) schedule on a carefully personal basis.

3. What is the rationale behind propensity score method? Explain? In reference to Lines 136-138, please elaborate what independent parameters were being referred here?

4. In reference to lines 165-168, discuss the role of preserving healthy tissues or healthy fringes surrounding the tumor in reference to https://doi.org/10.1016/j.cmpb.2020.105781.

An essential consequence of this safety trial is when a safe treatment of such a sensitive organ as the brain 166 with such advanced disease as GBM could be performed, we may also expect safety for various other organs.

5. There is no discussion on duration of mEHT? Also, insert a table to classify the disease conditions (grade III-IV) etc. Discuss whether it can be used standalone therapy or a combinatorial therapy. Tabulate the information specified in Line 186-188. Bevacizumab + FOLFOX? Similarly, for Rectal cancer? Paclitaxel and cisplatin chemotherapy for ovarian cancer? Tabulate the information? Temozolomide? (Bevacizumab + Boswellia serrata + Curcumin)?

6. Write all abbreviations next to the mention. For example: what is FIGO Stage III? NSCLC appeared in line 204 and explained in line 209 (non-small-cell lung cancer)/ what is pCR?

7. Why multiple single-arm studies are appropriate for comparison? Why the authors only use Kaplan-Meier probability comparison?

8. Discuss mEHT values for different cancers? Also, elaborate other thermal therapies. compare and contrast magnetic nanoparticle hyperthermia [https://doi.org/10.1115/1.4046967], photothermal therapies etc.?

9. Discuss the role of blood perfusion in mEHT? How blood perfusion is affected during modulated electro-hyperthermia? Discuss the heterogeneous blood perfusion distribution within tumor [http://hdl.handle.net/11603/25298] and discuss its variability between different subjects? Discuss blood perfusion during heating in relevance to https://doi.org/10.1016/j.icheatmasstransfer.2022.106046.

10. Lines 506-508

The match is probably so accurate because the GBM rarely gives metastases, so the distant metastases 507 do not modify the survival.

We disagree with such inference. Provide rationale to satisfy this claim.

11. Quantitative comparisons are missing and qualitative comparisons should be better explained. Revise Conclusion section.

We are looking forward to receive your revised Manuscript.

Avoid technical jargons in the Manuscript. Quantitative and qualitative comparisons are missing. Revise Introduction section and add context for mEHT.

Author Response

Answer to the second Reviewer

Thank you very much for your valuable comments. All your remarks, comments, requests are included and explained in the corrected text. Two extra points which we explain:

  1. The selectivity (targeting mostly the cancer cells avoid to hurt the normal) is explained in extension. The otherwise low overall power (150W, which is usually a fraction of the conventional hyperthermia heating) also ensures to keep the healthy environment intact. The glioma explanation is also extended.
  2. The comparison of the survival when the treatment has the same protocol, has similarity for the marketed medications, which after its approval expected having similar results worldwide.

The submission has altogether a considerable extension.

Thanks again for your critical proposals.

On behalf of the authors:

Marcell A. Szasz

Reviewer 3 Report

Modulated electro-hyperthermia (mEHT) has been used to treat different types of cancer, for example: cervical, glioblastoma, pancreatic, ovarian, lung, rectal and others. It would be important and illustrative to show a table including the type of cancer, number of patients if the therapy is with chemotherapy, radiotherapy or combination, including mEHT, and it would also be important to indicate in the table the maximum doses used, because it is confusing to talk about safety (Page 4 lines 159-175) if we talk about two types of cancer (glioblastoma vs. ovarian).

In survival time: survival periods are mentioned and the text is a bit confusing because they are types of cancer with particular conditions, in one case it is patients with cervical cancer and also HIV, or advanced cervical cancer, or recurrent cervical cancer and treated with chemotherapy, cases of non-small cell lung cancer and gastro intestinal cancer are also mentioned, I suggest that a table be drawn up and indicate in it the most important parameters of treatment with mEHT, including survival time.

Figure 1 shows the survival results from a pool of six mEHT studies, in which the diagnosis times and the probability of survival are observed, and it is not described how the curve marked as No. 7 is generated. And the results seem to indicate that the diagnosis time does not matter, since the probability of survival is similar.  I suggest that the table indicate that the studies correspond to cases of glioblastoma.

I also suggest including two recent studies where the effects of mEHT treatment on patient survival are clear (Modulated electrohyperthermia in locally advanced cervical cancer: Results of an observational study of 95 patients. Lee, Sun Young; Lee, Dong Hyun; Cho, Dong-Hyu. Medicine. 102(3):e32727, January 20, 2023. https://doi.org/10.1097%2FMD.0000000000032727

Meta-Analysis of Modulated Electro-Hyperthermia and Tumor Treating Fields in the Treatment of Glioblastomas. Attila Marcell Szasz, Elisabeth Estefanía Arrojo Alvarez, Giammaria Fiorentini, Magdolna Herold, Zoltan Herold, Donatella Sarti and Magdolna Dank. Cancers 2023, 15(3), 880; https://doi.org/10.3390/cancers15030880)

The discussion does not include elements that are important in the evaluation, such as the heterogeneous impedance properties of tumor cells, the high density of rafts in membranes, and the pathological (spatio-temporal) arrangements of tumor cells (see references [Andras Szasz, (2020) Towards the Immunogenic Hyperthermic Action: Modulated ElectroHyperthermia. Clinical Oncology and Research DOI: 10.31487/j.COR.2020.09.07; Andras Szasz (2021). The Capacitive Coupling Modalities for Oncological Hyperthermia Open Journal of Biophysics vol. 11. No. 3. DOI: 10.4236/ojbiphy.2021.113010].

Author Response

Answer to the third Reviewer

Thank you very much for your valuable comments. All your remarks, comments, requests are included and explained in the corrected text.

The submission has altogether a considerable extension.

Thanks again for your critical proposals.

On behalf of the authors:

Marcell A. Szasz

Round 2

Reviewer 2 Report

Lines 679-683 The selected energy absorption of rafts heats the TME and tissues to a lesser extent. The standard applied SAR in nanoparticles, considering their weight heating, is 0.1-1.5 MW/kg. The approximation of the absorbed power of rafts in selection is SAR>1 MW/kg, which is similar to the standard MNP energies.

The MNPs distribution is heterogeneous and furthermore they re-distribute during heating induced under the exposure of alternating magnetic fields. Magnetic nanoparticles migrate from the region of higher concentration to the region of lower concentration due to change of interstitial space (tumor microenvironment) [https://doi.org/10.1016/j.icheatmasstransfer.2021.105393]. Specific Absorption Rate of magnetic nanoparticles is recently expressed using microCT image guidance [https://doi.org/10.1016/j.ijthermalsci.2022.107996]. Include such recent developments under this comprehensive review.

Please proof-read your work.

Author Response

Dear Reviewer,

We had accepted all the comments and corrected the text. 

Please see the new submission according to the kind request of the Reviewers.